# Cost-effectiveness of mechanical thromboprophylaxis for cesarean deliveries in Brazil

**Alex Veloz**[1]*, **Ubong Silas**[2], **Rhodri Saunders**[2], **Jody Grisamore**[3‡], **André Luiz Malavasi**[4‡]

1 Health Economics and Clinical Outcomes Research, Xavier University, Cinncinati, Ohio, United States of America, 2 Coreva Scientific GmbH & Co. KG, Koenigswinter, Germany, 3 Labor and Delivery, Northwestern Medicine Prentice Women's Hospital, Chicago, Illinois, United States of America, 4 Faculty of Medicine, Universidade de São Paulo, São Paulo, Brazil

☯ These authors contributed equally to this work.
‡ JG and ALM also contributed equally to this work.
* peer-review@coreva-scientific.com

## Abstract

### Objective

To evaluate the cost-effectiveness of using mechanical thromboprophylaxis for patients undergoing a cesarean delivery in Brazil.

### Methods

A decision-analytic model built in TreeAge software was used to compare the cost and effectiveness of intermittent pneumatic compression to prophylaxis with low-molecular-weight heparin or no prophylaxis from the perspective of the hospital. Related adverse events were venous thromboembolism, minor bleeding, and major bleeding. Model data were sourced from peer-reviewed studies through a structured literature search. A willingness-to-pay threshold of R$15,000 per avoided adverse event was adopted. Scenario, one-way, and probabilistic sensitivity analyses were performed to evaluate the impact of uncertainties on the results.

### Results

The costs of care related to venous thromboembolism prophylaxis and associated adverse events ranged from R$914 for no prophylaxis to R$1,301 for low-molecular-weight heparin. With an incremental cost-effectiveness ratio of R$7,843 per adverse event avoided. Intermittent pneumatic compression was cost-effective compared to no prophylaxis. With lower costs and improved effectiveness, intermittent pneumatic compression dominated low-molecular-weight heparin. The probabilistic sensitivity analyses showed that the probability of being cost-effective was comparable for intermittent pneumatic compression and no prophylaxis, with low-molecular-weight heparin unlikely to be considered cost-effective (0.07).

**Data Availability Statement:** All relevant data are within the paper and its Supporting information files.

**Funding:** This work was funded by Cardinal Health https://www.cardinalhealth.com/en.html The

funders had no role in study design, data collection and analysis, decision to publish, or preparation of the manuscript.

**Competing interests:** I have read the journal's policy and the authors of this manuscript have the following competing interests:US is an employee and RS is the owner of Coreva Scientific GmbH & Co KG; along with AV, Coreva Scientific received consultancy fees for performing, analyzing, and communicating the work presented here. ALM is an obstetrician/gynecologist practicing in Brazil and a member of scientific advisory board for Bayer, Sanofi, Cardinal Health and Organon.

## Conclusions

Intermittent pneumatic compression could be a cost-effective option and is likely to be more appropriate than low-molecular-weight heparin when used for venous thromboembolism prophylaxis for cesarean delivery in Brazil. Use of thromboprophylaxis should be a risk-stratified, individualized approach.

## Introduction

Venous thromboembolism (VTE) is a disease condition that presents as a deep vein thrombosis (DVT) or in more severe cases, a pulmonary embolism (PE). It involves the development of blood clots (thrombi) inside the deep veins of the legs (DVT), and the potential movement of the thrombi to the pulmonary artery or its branches (PE) [1]. Bleeding-thromboembolism duality dilemma is a current challenge of obstetrics. Not long ago, the greatest concern of health care professionals was accidental hemorrhage during child birth, but today, this fear is compounded by VTE [2]. Pregnant individuals are at higher risk of experiencing a VTE event compared to non-pregnant populations, this is due to hypercoagulability, venous stasis, and endothelial injury [1, 3]. Consequently, embolism, according to research by the World Health Organization (WHO), is one of the main causes of global maternal death [2]. Covid-19 has been identified to have a potentially concerning thrombotic effect on pregnancy [4]. An increase in D-dimer, a protein associated with VTE, and a higher incidence of maternal vascular thrombosis have been reported in Covid-19 infected pregnant individuals compared to the non-infected group [5, 6]. Given this possible association, the International Society of Thrombosis and Hemostasis (ISTH) and Ministry of Health in Brazil recommends all pregnant individual receive a pharmacological thromboprophylaxis [7]. Additionally, birth through cesarean section has been identified as one of the leading risk factors for postpartum VTE events [1, 7]. Brazil is known to have the second highest cesarean section rates in the world after Dominican republic [8], with recent estimates that cesarean delivery accounted for 55.8% of all deliveries in the country between 2014 and 2017 [9].

To minimize the risk of VTE during cesarean delivery, it is the recommendation of several clinical guidelines that pregnant patients should receive thromboprophylaxis according to their VTE risk status [10–13]. More specifically, the Brazilian Federation of Gynecology and Obstetrics Associations (*Federação Brasileira das Associações de Ginecologia e Obstetrícia*, FEBRASGO) recommends the use of anticoagulants (LMWH) as the drug of choice for the prevention of VTE in pregnancy [13]. But all anticoagulants have inherent bleeding risks which can result in serious clinical and cost consequences. A meta-analysis showed an increased risk of postpartum hemorrhage [Relative Risk (RR) 1.52, 95% CI; 1.22–1.88] with the use of prophylactic low-molecular-weight heparin (LMWH) compared to placebo or no treatment in cesarean delivery [14]. In an Argentinian study, the average medical costs of bleeding events associated with VTE prophylaxis have been estimated to range from approximately A$27,269 [2021 Brazilian Real = R$4,028] for minor bleeding to AR$223,606 [2021 Brazilian Real = R$36,729] for major bleeding [15].

An alternative approach of preventing VTE without increasing the risk of bleeding is by using mechanical means to improve venous blood flow velocity in the lower limbs. Intermittent pneumatic compression (IPC) is an example of a mechanical thromboprophylaxis. The American College of Obstetricians and Gynecologists (ACOG) recommends the use of pneumatic compression devices for all women undergoing cesarean delivery [10, 16]. According to

a large registry study in the US, the use of IPC in all cesarean deliveries has seen a 85.95% reduction ($p = 0.038$) in postoperative PE death [17]. Furthermore, limb compression is also employed as a technique in reducing the risk of intraoperative hypotension during epidural anesthesia for cesarean delivery and has become the standard of care [18, 19]. According to a recent Cochrane review, leg compression during spinal anesthesia was shown to be an effective method to significantly reduce the risk of hypotension [RR 0.61, (95% CI; 0.47–0.78)] [20]. Therefore, using IPC during and after cesarean delivery might provide an added clinical advantage of reducing the risk intraoperative hypotension.

However, the absolute incidence of VTE after cesarean delivery is low. A large registry study of about 1.2 million cesarean sections reported an incidence of 0.21% [95% confidence interval (CI), 0.20–0.22%] between 2015 and 2017 in the United States [21]. Despite the low risk of VTE in cesarean delivery, the rate of VTE occurrence becomes significant when other risk factors are taken into consideration, risk factors such as history of VTE, thrombophilia, sickle cell disease, inflammatory bowel disease, cancer, obesity, preeclampsia and Covid-19 infection [7, 22]. Identification of at-risk individuals and subsequent selection of the most appropriate form of VTE prophylaxis is thus of paramount importance to providing cost-effective healthcare. Currently, there is no health economic comparison between IPC and standard of care during cesarean delivery in Brazil. Hence, there are open questions about the cost-effectiveness of the different prevention strategies in Brazilian hospitals. The primary objective of this analysis was to evaluate the cost-effectiveness of IPC compared to no prophylaxis or LMWH in patients undergoing a cesarean section.

## Methods

A cost-effectiveness analysis was performed from the perspective of a Brazilian hospital using a decision-tree health economic model. The patient population considered were pregnant patients undergoing a cesarean delivery in Brazil. The modeled time horizon of the analysis was from receiving the cesarean section to discharge from hospital, reported to be two to three days in Brazil [23]. Our findings in this publication are reported according to the *Consolidated Health Economic Evaluation Reporting Standards* (CHEERS 2022) checklist (see S1 Table in S1 File) [24].

### Patient population

Included in this cost-effectiveness analysis was a theoretical cohort of pregnant patients requiring a cesarean delivery. The model used clinical data extracted from peer-reviewed studies as inputs; no primary clinical data was processed, and the analysis did not involve the use of human subjects. No exclusion criteria were applied in the model, though readers should familiarize themselves with the primary literature cited to understand any inclusion and exclusion bias that may be present in the literature inputs used.

### Model structure

A decision-tree model was developed using TreeAge Pro® 2022 software (Healthcare version) to compare three different thromboprophylaxis strategies: mechanical prophylaxis (IPC), pharmacological prophylaxis (LMWH), and no prophylaxis (Fig 1). The events in the care pathway were assumed to be non-recursive and together with the short time horizon (three days) were the reason for adopting a decision-tree model structure [25]. Discounting of costs was deemed unnecessary due to the short time horizon.

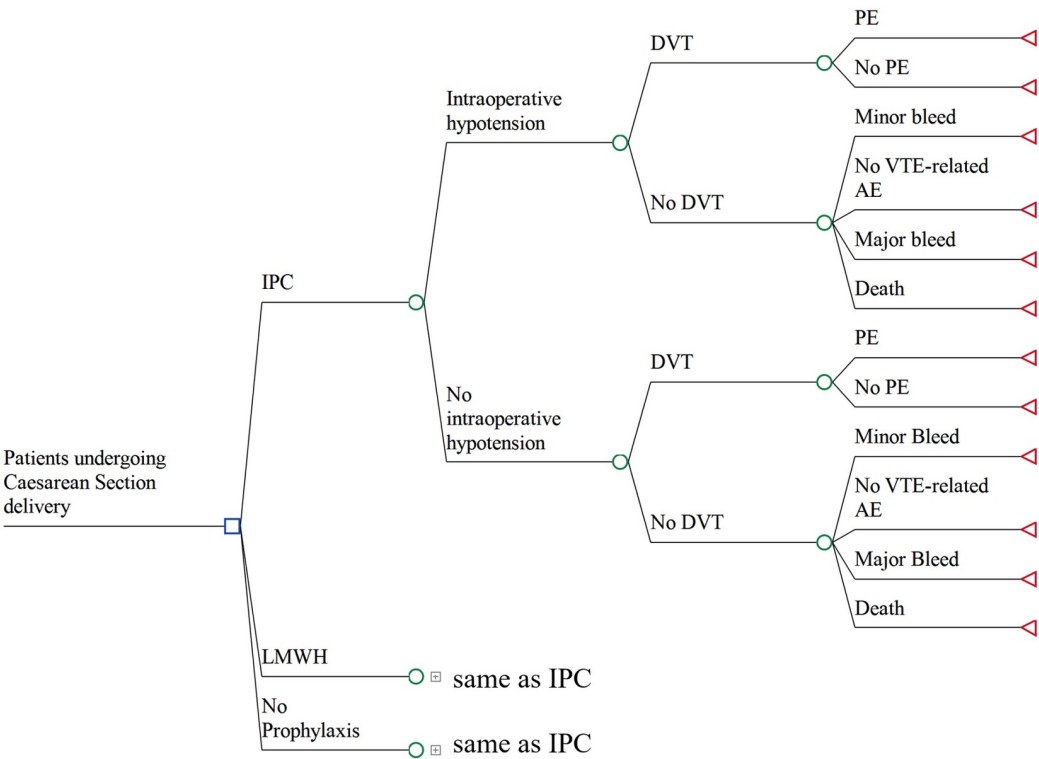

**Fig 1. Decision-tree model structure comparing three thromboprophylaxis strategies.** The pathways of LMWH and no prophylaxis strategies are the same as with IPC. Blue square node represents the initial decision node of each strategy; green circle nodes represent chance nodes; red triangle nodes represent terminal nodes. *Abbreviations*: *IPC, intermittent pneumatic compression; LMWH, low-molecular-weight heparin; DVT, deep vein thrombosis, AE, adverse event, PE, pulmonary embolism.*

## Literature search

A structured literature search was performed on PubMed to identify relevant peer-reviewed studies for informing the model design and providing the input data. The search was initially conducted in April 2022 and further targeted searches were carried out for any identified data gaps. Data from meta-analyses and randomized controlled trials, as well as most recent publications from Central and South America were favored.

## Clinical care pathway

The model simulates the clinical care pathway of a patient during and after cesarean delivery in the hospital. LMWH is the recommended thromboprophylaxis post cesarean section in Brazil and the no prophylaxis strategy was included because according to the FEBRASCO guideline on thromboprophylaxis, prophylaxis should be administered according to VTE and bleeding risk factors [13]. IPC can and is sometimes recommended to be used in the intraoperative setting (for prevention of intraoperative hypotension) as well as in the postoperative setting for VTE prophylaxis. In our model, intraoperative hypotension is not considered as a VTE or thromboprophylaxis-related adverse event, rather as an intermediate event highlighting the effect of intraoperative use of IPC.

## Clinical effectiveness

A patient in the model may or may not have intraoperative hypotension during the cesarean section and may or may not have any of the postoperative adverse events (i.e., minor bleeding, major bleeding, DVT and PE). A patient with VTE, assumed to be DVT initially, has a risk of developing a subsequent PE.

The primary efficacy outcome in this analysis was the combined incidence of all cesarean sections not impacted by any VTE-related adverse event. Whereby, the value one means that all VTE-related adverse events were avoided, and the value zero means that every cesarean section was impacted by at least one adverse event. The expected effectiveness of each strategy was determined by calculating the respective probabilities and summing them across all the pathways. The probabilities of the adverse events (DVT, PE, minor bleeding, and major bleeding) were extracted from a previously published cost-effectiveness analysis comparing thromboprophylaxis with IPC versus no prophylaxis (see Table 1) [26]. A Cochrane review of

**Table 1. Baseline clinical and cost input parameters for the model.**

| Parameters by measure | Value (Variance)[a] | Data source |
|---|---|---|
| **Incidences, %[e]** | | |
| DVT post cesarean delivery | 0.70 (0.10–3.0) | [26] |
| PE due to DVT | 20.0 (15–25) | [26] |
| Minor bleed | 6.80 (3–10) | [26] |
| Major bleed | 1.40 (0.60–2.0) | [26] |
| Mortality | 0.0026 | [29] |
| Hypotension | 66.29 | [20] |
| **Costs, R\$[b]** | | |
| DVT | 9,524 (7,486–12,036) | [15] |
| PE | 10,618 (7,666–13,557) | [15] |
| Minor bleed | 4,479 | [15] |
| Major bleed | 36,730 | [15] |
| Hypotension | 30 (20–40) | [30] |
| LMWH | 50 | [31] |
| IPC | 102 | Data on file[f] |
| **Relative efficacy, RR** | | |
| IPC versus LMWH on DVT or PE | 0.87 (0.08–9.50) | [32] |
| **LMWH versus no treatment on DVT** | 0.33 (0.01–7.93) | [33] |
| LMWH versus no treatment on minor bleeding | 2.12 (1.15–3.93) | [33] |
| LMWH versus no treatment on major bleeding | 1.48 (0.25–8.72) | [33] |
| Lower limb compression versus control on hypotension [d] | 0.36 (0.22–0.56)[c] | [34] |

Notes:

[a] 10% variation was assumed where variance was not provided;

[b] All cost data were inflated to 2021 Brazilian Real (R\$);

[c] RR was estimated from the reported odds ratio;

[d] Relative efficacy of IPC on hypotension was estimated from that of lower limb compression, the event rate of hypotension with IPC was then calculated at 66.29% x 0.36 (RR) = 23.86%;

[e] Baseline incidences with no prophylaxis,

[f] IPC cost data provided by Cardinal Health

**Abbreviations**: DVT, deep vein thrombosis; PE, pulmonary embolism; LMWH, low-molecular-weight heparin; RR, relative risk; IPC, intermittent pneumatic compression.

preventing adverse events during spinal anesthesia for cesarean delivery provided the probability of intraoperative hypotension [20].

## Costs

All costs were estimated from the hospital payer perspective, therefore, only direct hospital costs of thromboprophylaxis and costs of managing associated adverse events were considered. Delivery-related costs, including the cost for a cesarean delivery were not modeled as they were assumed to be equal for all comparators. Cost data were mostly extracted from published studies, and, when unavailable, experts were consulted. Costs from previous years and currencies were converted to 2021 Brazilian Real (R$) using the respective exchange rates and Brazil's Consumer Price Index [27, 28]. A summary of the all the cost data and their sources are presented in Table 1.

Cost per patient was calculated for each strategy. The incremental cost-effectiveness ratio (ICER), defined as the additional cost per adverse event avoided, was estimated for mechanical prophylaxis compared to no prophylaxis or pharmacological prophylaxis. The ICER is calculated as:

$$ICER = \frac{Cost\ of\ strategy\ 1 - Cost\ of\ strategy\ 2}{Effectiveness\ of\ strategy\ 1 - Effectiveness\ of\ strategy\ 2}$$

## Model assumptions

To estimate the cost-effectiveness of the three strategies over the time horizon of three days, the following assumptions were made:

- The postoperative adverse events were assumed to be mutually exclusive except for PE which evolves from a DVT event; a patient could experience only one of the four adverse events.

- As there is no consensus on an accepted willingness-to-pay threshold for Brazil, a previously reported willingness-to-pay value of R$15,000 per avoided AEs after thromboprophylaxis was used [35].

- The non-AE related mortality of patients within the time horizon was assumed to be zero.

- Only maternal outcomes were considered and no impact of the prophylaxis regimen on the infant was assumed.

- Long-term VTE-related adverse events such as chronic thromboembolic pulmonary hypertension (CTEPH) and post-thrombotic syndrome (PTS) were assumed not to present within the time horizon of this model. As such, they are unlikely to influence a hospital payers' determination of which method of VTE prophylaxis is optimal for the hospital to pursue.

## Sensitivity analysis

One-way sensitivity, probabilistic sensitivity, and scenario analyses were performed to explore the uncertainty around model outcomes. The aim of the one-way sensitivity analysis was to identify the model inputs that have the greatest impact on the model outputs; results are presented as a tornado diagram. The impact of the risk of VTE on per patient cost of each strategy was also explored. For the probabilistic sensitivity analyses, a distribution was assigned to each parameter based on the mean and their corresponding variance: lognormal distribution for relative risks, gamma distribution for cost data and beta distribution for transition probabilities. In our analysis, the distributions of each parameter were randomly sampled over 1,000

Monte Carlo simulations to determine the overall proportion at which each thromboprophylaxis is cost-effective at a willingness-to-pay (WTP) value. A WTP threshold of approximately R$15,000 per avoided AEs was considered.

The scenario analysis considered the cost per patient in hypothetical situations where IPC is used only during or only after cesarean delivery.

## Results

The cost per patient was estimated at R$914 with no prophylaxis. For the base case, the use of IPC and LMWH increased the cost to R$ 950 and R$1,301, respectively. For the effectiveness, the results of no prophylaxis and IPC prophylaxis were comparable (Table 2), with 91.2% and 91.6% of patients avoiding any adverse event. The effectiveness of LMWH was lower, with only 86.1% avoiding any adverse event. Given the lower cost and increased effectiveness associated with no prophylaxis and IPC prophylaxis, the use of these two strategies dominated the use of LMWH. The ICER for IPC prophylaxis versus "no prophylaxis" was calculated as R$7,843 per AEs avoided (Table 2), substantially below the WTP threshold (R$15,000 per avoided AEs).

According to the results of the one-way sensitivity analysis (see S1 and S2 Figs in S1 File), the cost-effectiveness result of IPC versus no prophylaxis was heavily influenced by the probability for DVT (estimate 0.03–0.001). On the other hand, the key drivers of the cost difference between IPC and no prophylaxis were the relative risks of LMWH on minor and major bleeding, the relative risk of IPC on DVT, and the probabilities of minor bleeding and major bleeding. This means that the cost-effectiveness of IPC versus LMWH was heavily influenced by bleeding events.

Considering only the incidence of VTE, a plot of its uncertainty estimates showed that the use of IPC is the more cost-saving strategy in comparison to no prophylaxis when the incidence of VTE is higher than 1.20% (Fig 2).

The results of the PSA were represented using a cost-effectiveness plane of 1,000 Monte Carlo simulations and a cost-effectiveness acceptability curve. Compared to no prophylaxis, IPC was superior in only 17.6% of the simulated cases and in 86.3% of cases compared to LMWH, which means that IPC has a higher likelihood of being cost effective in comparison to LMWH. The scatterplots in Fig 3 show the relationship between costs and effectiveness for all simulated strategy. At a WTP threshold of R$15,000 per VTE event avoided, the probability of being cost-effective was 0.49 for "no prophylaxis" 0.44 for IPC, and 0.07 for LMWH (Fig 4).

The impact of using IPC intraoperatively and postoperatively is clear from the scenario analyses (see Table 3). That is, if used only during and only after cesarean delivery, IPC is likely to have a R$54 and R$13 higher cost per patient than the baseline cost respectively where the use of IPC is modeled until discharge from the hospital. The effectiveness was also shown to be comparable among the different scenarios.

**Table 2. Model results.**

|  | Cost, R$ | Incremental Cost, R$ | Effectiveness | Incremental Effectiveness | ICER, R$ |
|---|---|---|---|---|---|
| No prophylaxis | 914 |  | 0.912 |  |  |
| IPC | 950 | 36 | 0.916 | 0.005 | 7,843 |
| LMWH | 1,301 | 351 | 0.861 | -0.055 | Dominated |

**Abbreviations**: R$, Brazilian Real; IPC, intermittent pneumatic compression; LMWH, low-molecular-weight heparin; ICER, incremental cost-effectiveness ratio

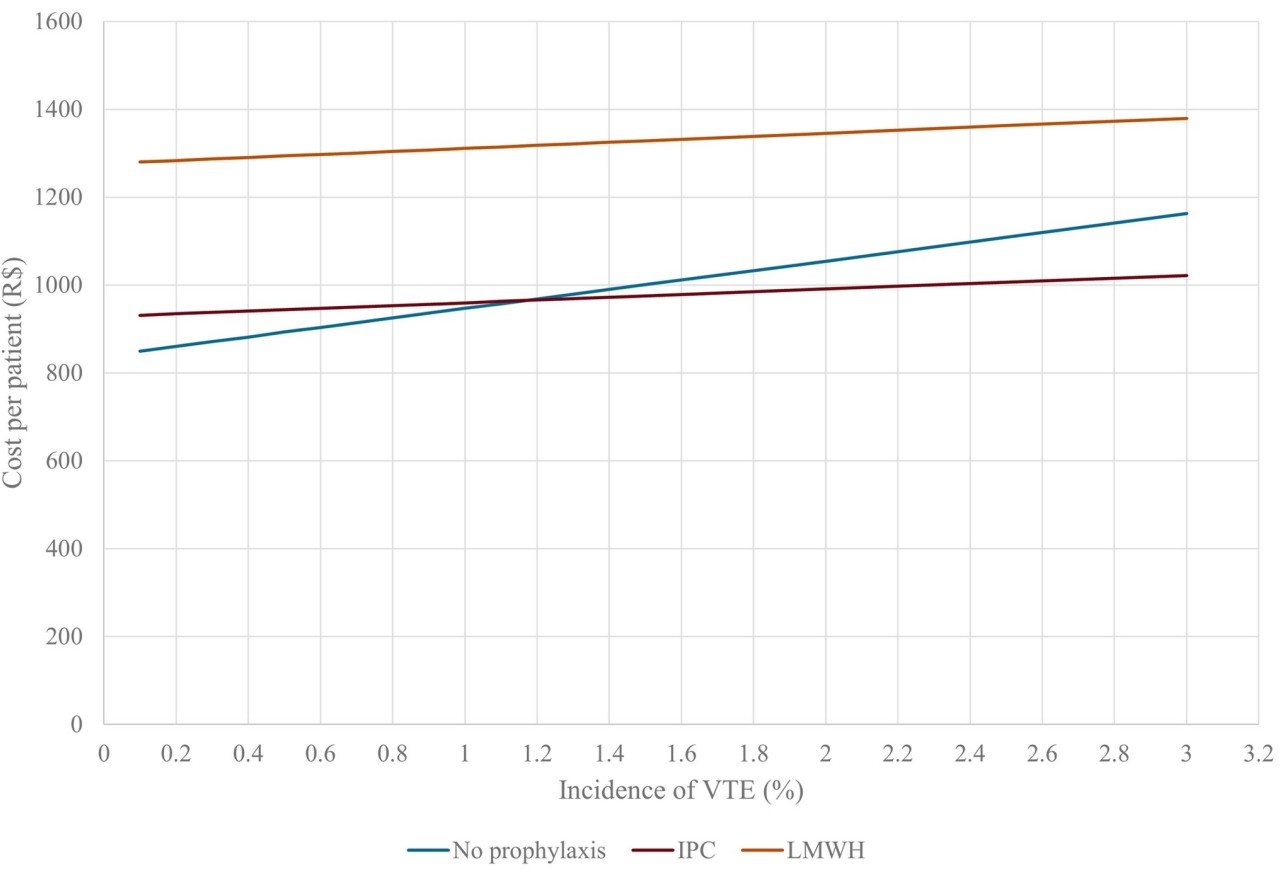

**Fig 2. Illustrating the cost per patient of the three strategies with increasing incidence of VTE. Abbreviations**: R$, Brazilian Real; IPC, intermittent pneumatic compression; LMWH, low-molecular-weight heparin.

## Discussion

A current problem in obstetric practice is VTE, especially given the reductions in hemorrhagic complications and infections during pregnancy and puerperium observed in more developed settings. Mechanical and pharmaceutical preventative measures have been used to reduce the incidence of VTE and its immediate and long-term effects [36].

 This cost-effectiveness study analyzed the cost-effectiveness of using a mechanical thromboprophylaxis compared to no prophylaxis or LMWH during cesarean delivery from a hospital perspective in Brazil. The result of our analysis showed that the cost of the prophylaxis strategies ranged from R$914- R$1,301 per patient. Our analysis also showed that IPC could be a cost-effective option compared to no prophylaxis as the calculated ICER of R$7,843 was lower than the estimated willingness-to-pay threshold of R$15,000 per avoided AEs. Furthermore, the PSA result showed that at the WTP threshold, probability of cost-effectiveness of no prophylaxis and IPC were comparable. Compared to LMWH however, the difference was substantial. With an ICER of -R$6,389, our model showed that the use of IPC during cesarean delivery avoided complications at a lower cost of hospital care. This may be due to the inherent risk of bleeding caused by administering LMWH, an adverse event which significantly increases the overall cost of complications. Although recommended by the Brazilian Federation of Gynecology and Obstetrics Associations (FEBRASCO) as the thromboprophylaxis of

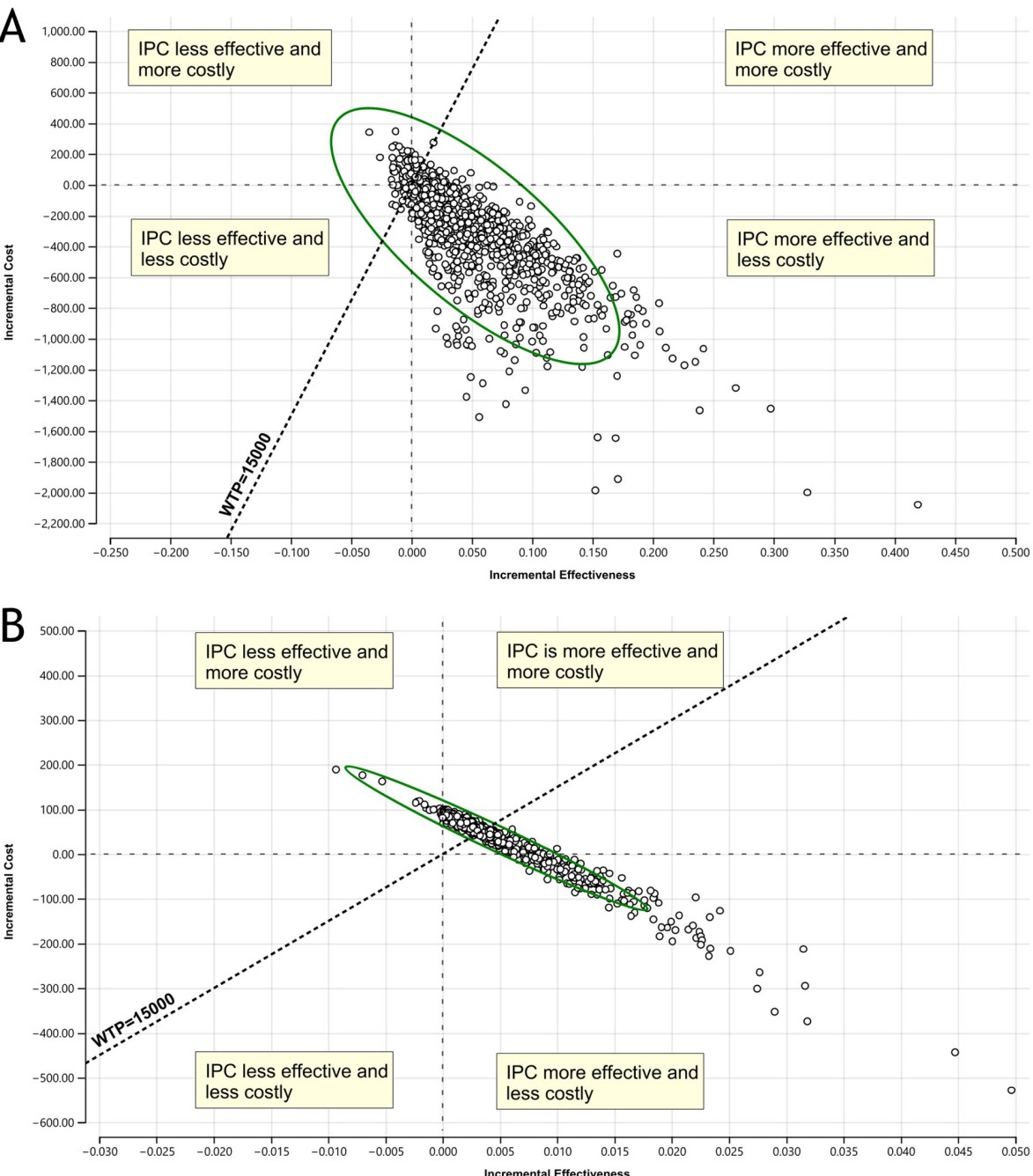

**Fig 3. Cost-effectiveness plane showing the scattered plots of 1000 Monte Carlo simulated cost-effectiveness of IPC versus LMWH (A) and IPC versus No prophylaxis (B).** Each point in the graph represents a single simulated model calculation by plotting the incremental cost (x-axis) and incremental effectiveness (y-axis). The WTP line splits the graph into points that favor the comparator strategy—LMWH/ no prophylaxis (below/right of line) and those that favor the baseline strategy—IPC (above/left of line). The green ellipsis shows the 95% confidence interval. **Abbreviations**: IPC, intermittent pneumatic compression; WTP, willingness-to-pay; LMWH, low-molecular-weight heparin.

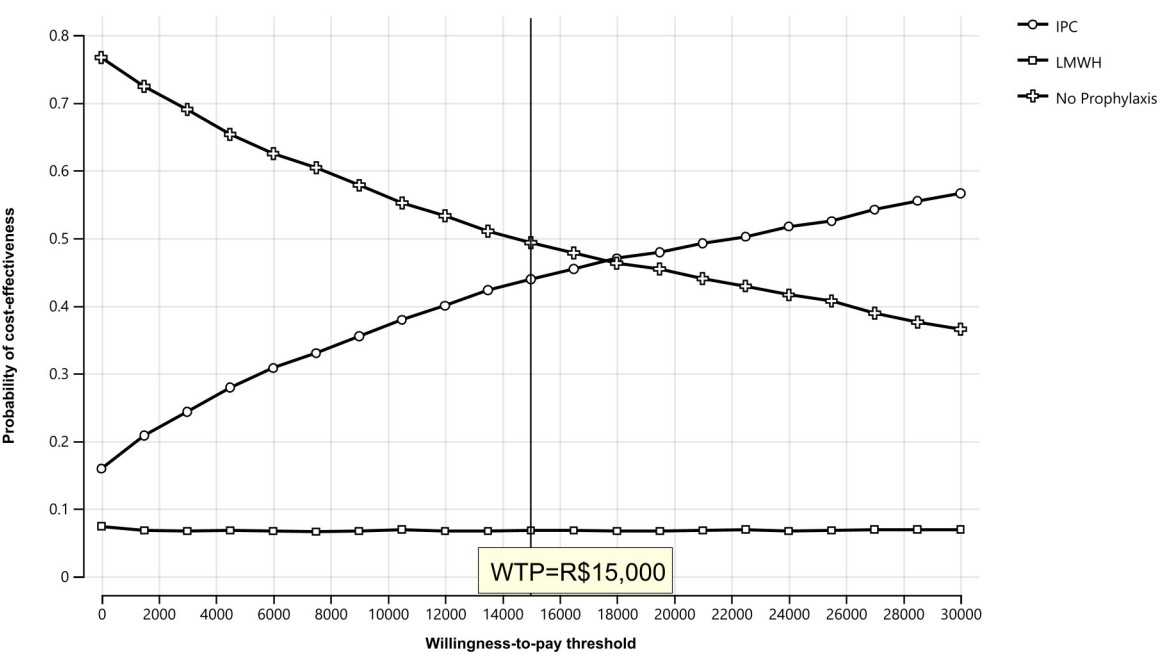

**Fig 4.** Cost-effectiveness acceptability curve showing the likelihood of cost-effectiveness of the three strategies: *Abbreviations*: *R$, Brazilian Real; IPC, intermittent pneumatic compression; LMWH, low-molecular-weight heparin; WTP, willingness-to-pay.*

choice, LMWH may not be the most cost-effective option available for the prevention of VTE during cesarean delivery as shown by the results of our analysis [13].

Currently, there is insufficient evidence supporting the cost-effectiveness of mechanical thromboprophylaxis in obstetrics and gynecology. In orthopedics, however, the use of IPC alone or in combination with an anticoagulant have been reported as cost-effective strategies in the United States and Australia [37, 38]. The cost-effectiveness of mechanical thromboprophylaxis in gynecology was estimated by Casele and colleagues in 2006 [26]. In this study, a decision-tree with a Markov model was used to compare the cost and the quality adjusted life years (QALYs) of IPC versus no prophylaxis at the time of cesarean delivery from a healthcare system perspective [26]. Unlike in our study, the cost and consequences of long-term AEs such as post thrombotic syndrome and cardiovascular accident were also modeled alongside the VTE and bleeding events [26]. The ICER of routine use of a mechanical thromboprophylaxis during cesarean delivery was estimated at $39,545 per QALY [26]. Again, the results of this study are not fully comparable to our results because of the differences in the model setup and the included adverse events. Despite these differences, the results of both analyses support the cost-effectiveness of IPC use for cesarean delivery.

Generally, the risk of VTE is known to be increased after delivery, a risk further compounded when the delivery is via a cesarean section [7]. In Brazil, where there is a high

**Table 3. Results of scenario analyses.**

| Scenario comparison | Cost, R$ | Effectiveness |
|---|---|---|
| **IPC—intra + post op** | 950 | 0.916 |
| **IPC—post op** | 963 | 0.916 |
| **IPC—intra op** | 1,004 | 0.912 |

proportion of cesarean deliveries, efforts are made to keep the risk of VTE at their lowest. For example, a VTE risk assessment model was introduced to designate the necessity and duration of thromboprophylaxis use. This has resulted in lower PE-related maternal death and an establishment of a risk stratification score [7]. However, the risk stratification and recommendations of the appropriate thromboprophylaxis are ongoing debates even among the different international guidelines [16]. Patients with a high risk of bleeding should avoid the use of anticoagulants such as LMWH, while patients with high risk of VTE without a high bleeding risk should be provided with the most cost-effective prophylaxis regimen. The results of our sensitivity analysis support the hypothesis that IPC would be a cost-effective option for cesarean delivery patients with high risk of VTE in Brazil, however, a risk-stratified individualized approach is recommended owing to the low absolute risk of VTE complications in obstetrics. The increasing rate of cesarean deliveries has not been justified by reduction in relevant clinical outcomes, and as indicated by Venturella et al. 2018, this trend may rather be linked to improper clinical practices and healthcare deficiencies [39]. Therefore, avoiding unnecessary cesarean deliveries could also reduce healthcare costs related to VTE in obstetrics practice in Brazil. A well-designed study comparing the clinical effectiveness of mechanical and pharmacological thromboprophylaxis could establish relative efficacy in this patient population and bring added validity to this model.

Our cost-effectiveness analysis has several limitations. As with any economic analysis, the results of the model are dependent upon the variables that are used. The relative efficacy of the two interventions, values that could materially alter the results, were derived from non-obstetric literature. The generalizability of its efficacy to an obstetric population is not well known. There is no direct comparison for mechanical and pharmacological VTE prophylaxis available for obstetrics and gynecology in Brazil. Relative efficacy comparing the two strategies was sourced from a study involving abdominoplasty surgical patients. Other data was sourced from international randomized control trials, which do not necessarily represent the Brazilian healthcare system. It is assumed that the efficacy and safety reported in those trials is comparable for all populations. Similarly, local cost data was not available for all inputs. The application of our study's results will depend on local clinical practices, such as whether VTE prophylaxis is routinely employed or if the standard of care involves no prophylaxis. Although our analysis involved the use of a theoretical cohort of patients and not human subject, our findings will support healthcare providers in determining the most cost-effective approaches for VTE prophylaxis when the intervention is deemed necessary.

## Conclusion

From the perspective of the hospital payer, the results of our analysis showed that the use of mechanical thromboprophylaxis could be a cost-effective alternative for preventing VTE-related complications in patients undergoing cesarean delivery in Brazil. For patients requiring thromboprophylaxis, providers should consider intra- and postoperative use of IPC.

## Supporting information

**S1 File.**
(DOCX)

## Acknowledgments

We thank everyone that contributed to the research and the creation of this manuscript.

## Author Contributions

**Conceptualization:** Alex Veloz, Ubong Silas, Rhodri Saunders, Jody Grisamore, André Luiz Malavasi.

**Formal analysis:** Alex Veloz, Ubong Silas, Rhodri Saunders.

**Investigation:** André Luiz Malavasi.

**Methodology:** Alex Veloz, Ubong Silas, Rhodri Saunders, André Luiz Malavasi.

**Visualization:** Alex Veloz, Ubong Silas, Rhodri Saunders.

**Writing – original draft:** Alex Veloz, Ubong Silas.

**Writing – review & editing:** Alex Veloz, Ubong Silas, Rhodri Saunders, Jody Grisamore, André Luiz Malavasi.

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
