## [Decision Letter · Decision Letter 0]

25 Apr 2023

PONE-D-23-07556Cost-effectiveness of mechanical thromboprophylaxis for cesarean deliveries in BrazilPLOS ONE

Dear Dr. Veloz,

Thank you for submitting your manuscript to PLOS ONE. After careful consideration, we feel that it has merit but does not fully meet PLOS ONE’s publication criteria as it currently stands. Therefore, we invite you to submit a revised version of the manuscript that addresses the points raised during the review process.

We look forward to receiving your revised manuscript.

Kind regards,

Antonio Simone Laganà, M.D., Ph.D.

Academic Editor

PLOS ONE

Journal Requirements:

Additional Editor Comments (if provided):

The topic of the manuscript is interesting. Nevertheless, the reviewers raised several concerns: considering this point, I invite authors to perform the required major revisions.

Reviewers' comments:

Reviewer's Responses to Questions

**Comments to the Author**

1. Is the manuscript technically sound, and do the data support the conclusions?

Reviewer #1: Yes

Reviewer #2: Yes

Reviewer #3: Yes

Reviewer #4: Partly

2. Has the statistical analysis been performed appropriately and rigorously? 

Reviewer #1: Yes

Reviewer #2: N/A

Reviewer #3: Yes

Reviewer #4: No

3. Have the authors made all data underlying the findings in their manuscript fully available?

Reviewer #1: Yes

Reviewer #2: Yes

Reviewer #3: No

Reviewer #4: Yes

4. Is the manuscript presented in an intelligible fashion and written in standard English?

Reviewer #1: Yes

Reviewer #2: Yes

Reviewer #3: Yes

Reviewer #4: Yes

5. Review Comments to the Author

Reviewer #1: This is a paper that aim to demonstrate the cost-effectiveness of using a mechanical

thromboprophylaxis compared to no prophylaxis or LMWH during cesarean delivery in Brazil

the topic is relevant in the field, actual according to the everyday increasing rate of CS worldwide, Brazil moreover, as authors state is among the countries with the highest rate of CS,

the topic is original, due to a lack of evaluation in the available literature of cost effectiveness of such procedure in obstetrics and gynecology

I would like to suggest authors within the discussion to mention, at least briefly even just a sentence regarding the need to avoid unnecessary cesarean section, as this will be the most coast effective strategy to reduce thrombo embolism after birth, many of the CS we perform world wide have not even an obstetric indication(with this purpose I would suggest to cite PMID: 29560505, doi: 10.36129/jog.2022.20

references are appropriate

I would suggest also to ameliorate the images definitions and resolution

best regards

Reviewer #2: I read with great interest the Manuscript titled "Cost-effectiveness of mechanical thromboprophylaxis for cesarean deliveries in Brazil", which falls within the aim of this Journal.

In my honest opinion, the topic is interesting enough to attract the readers’ attention. Methodology is accurate and conclusions are supported by the data analysis. Nevertheless, authors should clarify some point and improve the discussion citing relevant and novel key articles about the topic.

Authors should consider the following recommendations:

- Inclusion/exclusion criteria should be better clarified.

- The Authors did not mention the sample size calculation for their study. It is essential to specify this data in order to guarantee an adequate significance of the results obtained by the Authors.

-The authors have not adequately highlighted the strengths and limitations of their study. I suggest clarifying these points.

- Was this study registered? I could not find any information about this point.

- I could not find any information regarding the approval of the Institutional Review Board. Did author this approval before the study start?

- I could not find any information regarding the informed consent of enrolled patients. Did author obtain informed consent for each patient? Conversely, this point may raise serious concern from the ethical point of view.

- I recommend to highlight, at least briefly, the higher thrombotic risk in case of pregnant women with covid-19 infection (authors may refer to: PMID: 32975205; PMID: 36143264)

Reviewer #3: Thank you for the opportunity to review this paper. The paper is well-structured and easy to understand, with a primary focus on evaluating the cost-effectiveness of using mechanical thromboprophylaxis for patients undergoing cesarean delivery in Brazil. However, I have one major issue and several minor suggestions of the manuscript.

Major issue:

1. In decision tree models, it is recommended to avoid multiple outcomes being presented side by side. For example, outcomes should be presented as DVT/no DVT, with 'no DVT' then being further divided into 'major bleeding/minor bleeding'.

Minor suggestions:

2. References should be provided lines 74-77 recommendations for Brazilian Federation of Gynecology and Obstetrics Associations

3. Please ensure consistency in the phrasing used in lines 79 and 96 for RR and 95% CI.

4. The introduction in the last paragraph discusses how the absolute risk of VTE is low, but becomes significant when considering risk factors. However, the authors did not mention what these risk factors are, nor did they provide data to indicate how important they are.

5. Lines 303-315 should be simplified or incorporated into the introduction.

6. Please remove the sentence 'Due to the low absolute risk of VTE complications in obstetrics, thromboprophylaxis should be considered through a risk-stratified individualized approach' as it is not a conclusion drawn by the authors based on their research.

7. Figure 3A and 3B are not clear. I would suggest the authors to merge Figure 3A and 3B using different color or shape.

8. Although the authors claim that the results of their one-way sensitivity analysis are presented in the supplementary material, I was unable to locate them.

9. The authors are missing the following relative efficacy values in Table 1: 'IPC versus LMWH on minor bleeding/major bleeding' and 'LMWH versus no treatment on DVT/PE'.

Reviewer #4: In this cost-effectiveness analysis, the authors compare no prophylaxis to LMWH to IPC use alone in the post-partum c section population in Brazil. In their results, they report that IPC use alone is the most cost-effective strategy, contrary to current societal recommendations for the use of LMWH. I have several significant concerns about the design of the model that need to be addressed prior to publication.

Major

*The composite outcome of any adverse event creates a false equivalence between the impact of post-partum hemorrhage, DVT, and PE. These events have different short and long-term implications for patients, and so I find the use of it problematic as the long-term consequences of DVT and PE, such as PTS and CTEPH, are not accounted for and major and minor bleeding are weighted equally in spite of having very different implications. In the discussion the authors note that a similar study in gynecologic patients (ref 23) did include long-term sequelae such as PTS and CTEPH and also was done from a systems perspective rather than hospital. These differences that they cite make that study stronger and should be considered for inclusion in this model to strengthen the conclusions.

*Please provide sources for the costs of LMWH and IPC. "Expert opinion" is not an acceptable source for a cost (doi 10.1097/XCS.0000000000000534).

*Please state an explicit time horizon, rather than "a few" days.

*Is hypotension considered an adverse event? Please be more explicit about how hypotension is considered.

Minor

*The second sentence of the introduction is redundant and should already be known by the readership, and thus should be deleted (lines 58-60, page 3).

*Please edit for grammar and punctuation.

*Line 176, page 6: please correct spelling in ICER formula.

*Why is hypotension a branch point in the decision tree? There are no different event probabilities cited based on the presence of hypotension. Please clarify.

6. PLOS authors have the option to publish the peer review history of their article (what does this mean?). If published, this will include your full peer review and any attached files.

Reviewer #1: No

Reviewer #2: No

Reviewer #3: No

Reviewer #4: **Yes: **Kristina Nicholson

---

## [Author Response · Author response to Decision Letter 0]

1 Jun 2023

Reviewer #1:

1. This is a paper that aim to demonstrate the cost-effectiveness of using a mechanical thromboprophylaxis compared to no prophylaxis or LMWH during cesarean delivery in Brazil the topic is relevant in the field, actual according to the everyday increasing rate of CS worldwide, Brazil moreover, as authors state is among the countries with the highest rate of CS, the topic is original, due to a lack of evaluation in the available literature of cost effectiveness of such procedure in obstetrics and gynecology.

Response: We appreciate the Reviewer’s acknowledgment of the significance of our study's rationale. This support reinforces our confidence in the study’s objectives.

2. I would like to suggest authors within the discussion to mention, at least briefly even just a sentence regarding the need to avoid unnecessary cesarean section, as this will be the most cost effective strategy to reduce thrombo embolism after birth, many of the CS we perform worldwide have not even an obstetric indication(with this purpose I would suggest to cite PMID: 29560505, doi: 10.36129/jog.2022.20 references are appropriate

Response: Thank you for this suggestion and for providing the references. We acknowledge that avoiding unnecessary cesarean deliveries should in general reduce the risk of venous thromboembolism. To highlight this, the following text has been added to lines 306-310 in the discussion section of the manuscript:

“The increasing rate of cesarean deliveries has not been justified by a reduction in relevant clinical outcomes, and as indicated by Venturella et al. 2018, this trend may rather be linked to improper clinical practices and healthcare deficiencies.[39] Therefore, avoiding unnecessary cesarean deliveries could also reduce healthcare costs related to VTE in obstetrics practice in Brazil.”

3. I would suggest also to ameliorate the images definitions and resolution

Response: We thank the reviewer for bringing this to our attention and the quality of all the figures has been improved.

Reviewer #2:

I read with great interest the Manuscript titled "Cost-effectiveness of mechanical thromboprophylaxis for cesarean deliveries in Brazil", which falls within the aim of this Journal. In my honest opinion, the topic is interesting enough to attract the readers’ attention. Methodology is accurate and conclusions are supported by the data analysis. Nevertheless, authors should clarify some point and improve the discussion citing relevant and novel key articles about the topic. Authors should consider the following recommendations:

1. Inclusion/exclusion criteria should be better clarified.

Response: Thank you for identifying this lack of clarity.

The following text has been added to lines 105-111 to address the patient population of interest in our analysis:

“Patient population

Included in this cost-effectiveness analysis was a theoretical cohort of pregnant patients requiring a cesarean delivery. The model used clinical data extracted from peer-reviewed studies as inputs; no primary clinical data was processed, and the analysis did not involve the use of human subjects. No exclusion criteria were applied in the model, though readers should familiarize themselves with the primary literature cited to understand any inclusion and exclusion bias that may be present in the literature inputs used.”

2. The Authors did not mention the sample size calculation for their study. It is essential to specify this data in order to guarantee an adequate significance of the results obtained by the Authors.

Response: While we appreciate the reviewer’s feedback, we wish to point out that the analysis carried out in our paper uses a health economic modeling approach, hence does not require a sample size calculation to guarantee an adequate significance of the results. This has been clarified in the original manuscript in line, but maybe we have not made this sufficiently clear. Therefore, we have now stressed this item earlier in lines 99-100:

“A cost-effectiveness analysis was performed from the perspective of a Brazilian hospital using a decision-tree health economic model”

3. The authors have not adequately highlighted the strengths and limitations of their study. I suggest clarifying these points.

Response: Thank you, we hope to have adequately addressed the strengths and limitations of the study in the appropriate manner now (see lines 313 -330) 

4. Was this study registered? I could not find any information about this point.

Response: Thank you for highlighting this important point. Our study was not registered. To the best of our knowledge and according to the Consolidated Health Economic Evaluation Reporting Standards (CHEERS) best practice guidance, the registration of a health economic study is not mandatory.

5. I could not find any information regarding the approval of the Institutional Review Board. Did author this approval before the study start?

Response: The analysis in our study used a health-economic model with data from already published peer-reviewed studies. Therefore, our research did not involve the use of human subjects and did not warrant the approval of an Institutional Review Board.

6. I could not find any information regarding the informed consent of enrolled patients. Did author obtain informed consent for each patient? Conversely, this point may raise serious concern from the ethical point of view.

Response: Our research did not involve the use of human subjects, therefore did not involve the enrollment of patients and the signing of consent forms.

7. I recommend to highlight, at least briefly, the higher thrombotic risk in case of pregnant women with covid-19 infection (authors may refer to: PMID: 32975205; PMID: 36143264)

Response: We think this is an excellent suggestion and thank you for the references. We have added the following text to highlight the associated risk of VTE in pregnant women with Covid-19 infection and some recommendations (see lines 52-57):

“Covid-19 has been identified to have a potentially concerning thrombotic effect on pregnancy.[4] An increase in D-dimer, a protein associated with VTE, and a higher incidence of maternal vascular thrombosis have been reported in Covid-19 infected pregnant individuals compared to the non-infected group.[5, 6] Given this possible association, the International Society of Thrombosis and Hemostasis (ISTH) and Ministry of Health in Brazil recommends all pregnant individual receive a pharmacological thromboprophylaxis.[7]”

Reviewer #3:

Major issue: 

1. In decision tree models, it is recommended to avoid multiple outcomes being presented side by side. For example, outcomes should be presented as DVT/no DVT, with 'no DVT' then being further divided into 'major bleeding/minor bleeding'.

Response: Many thanks for bringing this to our attention. The model design has been adjusted accordingly. See Figure 1. Please note that due to the change in the model design, there has been a slight change to the model results and results of the sensitivity analysis. However, the conclusion and the key message remain the same.

Minor suggestions: 

2. References should be provided lines 74-77 recommendations for Brazilian Federation of Gynecology and Obstetrics Associations

Response: We apologize for this omission, the reference for the recommendation by the “Brazilian Federation of Gynecology and Obstetrics Associations” has now been included.

3. Please ensure consistency in the phrasing used in lines 79 and 96 for RR and 95% CI.

Response: Thank you for pointing out this inconsistency. This has now been amended. (See line 83)

4. The introduction in the last paragraph discusses how the absolute risk of VTE is low, but becomes significant when considering risk factors. However, the authors did not mention what these risk factors are, nor did they provide data to indicate how important they are.

Response: Thank you for pointing this out. The reviewer is right, outlining the risk factors in question gives a more balanced argument and well-informed hypotheses. Therefore, we have amended the texts in lines 88-91 and included the relevant references. It now reads:

“Despite the low risk of VTE in cesarean delivery, the rate of VTE occurrence becomes significant when other risk factors are taken into consideration, risk factors such as history of VTE, thrombophilia, sickle cell disease, inflammatory bowel disease, cancer, obesity, preeclampsia, and Covid-19 infection.[7, 22]”

5. Lines 303-315 should be simplified or incorporated into the introduction.

Response: Thank you for the recommendation. We made the decision to streamline the texts. The revised version now reads as follows (lines 286-299):

“Generally, the risk of VTE is known to be increased after delivery, a risk further compounded when the delivery is via a cesarean section.[7] In Brazil, where there is a high proportion of cesarean deliveries, efforts are made to keep the risk of VTE at their lowest. For example, a VTE risk assessment model was introduced to designate the necessity and duration of thromboprophylaxis use. This has resulted in lower PE-related maternal death and an establishment of a risk stratification score.[7] However, the risk stratification and recommendations of the appropriate thromboprophylaxis are ongoing debates even among the different international guidelines.[16]”

6. Please remove the sentence 'Due to the low absolute risk of VTE complications in obstetrics, thromboprophylaxis should be considered through a risk-stratified individualized approach' as it is not a conclusion drawn by the authors based on their research.

Response: We agree with Reviewer, the sentence was not drawn from our analysis, it was rather as a recommendation. Sentence has been removed from conclusion.

7. Figure 3A and 3B are not clear. I would suggest the authors to merge Figure 3A and 3B using different color or shape.

Response: We have enhanced the qualities of Figure 3A and 3B and have merged both to into one figure as suggested. Thank you! 

8. Although the authors claim that the results of their one-way sensitivity analysis are presented in the supplementary material, I was unable to locate them.

Response: Thank you for the observation. The results of the one-way sensitivity analysis are presented as Tornado diagram in the supplementary material, but maybe we have not made this sufficiently clear. We have now changed the Figure Legends from “Tornado diagram analysis of…” to “One-way sensitivity analysis of…” to address this.

9. The authors are missing the following relative efficacy values in Table 1: 'IPC versus LMWH on minor bleeding/major bleeding' and 'LMWH versus no treatment on DVT/PE'.

Response: Thank you for pointing this out. It was an oversight on our part. The data and reference for relative efficacy of “LMWH versus no treatment on DVT” has now been added to Table 1. Unfortunately, there is no published data on the relative efficacy of IPC versus LMWH on minor/major bleeding in obstetrics and gynecology, therefore, a relative risk of 1 was assumed, that is no difference between IPC and no prophylaxis on minor/bleeding. However, the range of 0.9 -1.1 was tested in the sensitivity analysis of the results.

Reviewer #4:

In this cost-effectiveness analysis, the authors compare no prophylaxis to LMWH to IPC use alone in the post-partum c section population in Brazil. In their results, they report that IPC use alone is the most cost-effective strategy, contrary to current societal recommendations for the use of LMWH. I have several significant concerns about the design of the model that need to be addressed prior to publication.

Major 

1. The composite outcome of any adverse event creates a false equivalence between the impact of post-partum hemorrhage, DVT, and PE. These events have different short and long-term implications for patients, and so I find the use of it problematic as the long-term consequences of DVT and PE, such as PTS and CTEPH, are not accounted for and major and minor bleeding are weighted equally in spite of having very different implications. In the discussion the authors note that a similar study in gynecologic patients (ref 23) did include long-term sequelae such as PTS and CTEPH and also was done from a systems perspective rather than hospital. These differences that they cite make that study stronger and should be considered for inclusion in this model to strengthen the conclusions.

Response: Thank you for the review of the model design. While we agree with the reviewer that bleeding, DVT and PE have long term implications for the patients, we would like to focus on the fact that our analysis was carried out from a hospital perspective. Although healthcare systems often look at a societal perspective, particularly for pharmaceutical purchases, this is not always the case for medical devices. Devices generally have no additional reimbursement, and their costs are accounted for in DRG payments for the procedure that the patient undergoes, in this case C-section. This means that the hospital must determine whether the purchase of IPC devices and their use in C-section makes economic and clinical sense in a time frame that is usually up to three days. If IPC is already used intraoperatively, then the question is whether maintaining its use to three days makes sense. As such, the model only accounts for the related costs and outcomes within this time frame. Long term adverse events such as chronic thromboembolic pulmonary hypertension (CTEPH) and post-thrombotic syndrome (PTS) are unlikely to be present within these three days, hence, our rationale for not including them in the model. Also, the costs of managing these long-term adverse events/complications are not covered by the hospital’s budget, therefore, outside the scope of our analysis. The perspective of the health economic evaluation is also the key difference between our analysis and the Casele et al., 2006 study. To further emphasize this, we have added the following text under model assumption (line 187-190):

“Long-term VTE-related adverse events such as chronic thromboembolic pulmonary hypertension (CTEPH) and post-thrombotic syndrome (PTS) were assumed not to present within the time horizon of this model. As such, they are unlikely to influence a hospital payers’ determination of which method of VTE prophylaxis is optimal for the hospital to pursue.” 

Finally, please note that although minor and major bleed are presented side-by-side in the model diagram, they are not weighted equally. They both have different incidences and costs (see Table 1)

2. Please provide sources for the costs of LMWH and IPC. "Expert opinion" is not an acceptable source for a cost (doi 10.1097/XCS.0000000000000534).

Response: As pointed out by the reviewer, we have now provided a published reference for the cost of LMWH. The cost of IPC was provided by Cardinal Health (the manufacturer of IPC), a note explaining this point has been added to the manuscript (see line 198-199). These changes to the manuscript can be found in Table 1 “Baseline clinical and cost input parameters for the model”. 

3. Please state an explicit time horizon, rather than "a few" days.

Response: The time horizon used for the model was three days. Accordingly, we have made this clear in the “Method structure” (Line 116) and “Model assumptions” (Line 176) sections of the manuscript.

4. Is hypotension considered an adverse event? Please be more explicit about how hypotension is considered.

Response: No, in our model, hypotension is not considered as an adverse event but as an intermediate event that can occur during cesarean delivery. As IPC is sometimes used intraoperatively to reduce the risk of intraoperative hypotension, the branching of hypotension allows the continued use of IPC postoperatively for VTE prophylaxis to be explored in the correct clinical context. To address this feedback, we have updated the Decision-tree model structure, changing the wording from “Delivery w/o complications” to “No VTE-related adverse event” (See Figure 1). The following the text have also been added to the methods (Line 138-140):

“In our model, intraoperative hypotension is not considered as a VTE or thromboprophylaxis-related adverse event, rather as an intermediate event highlighting the effect of intraoperative use of IPC”

Minor 

5. The second sentence of the introduction is redundant and should already be known by the readership, and thus should be deleted (lines 58-60, page 3).

Response: Thank you for your suggestion. We agree that the sentence “It involves the development of blood clots (thrombi) inside the deep veins of the legs (DVT), and the potential movement of the thrombi to the pulmonary artery or its branches (PE)” may be redundant to certain readership, for example, medical professionals in the field of obstetrics and gynecology but to other readership of PLOS ONE journal, for example, health economists, this sentence may provide a bit of background context on the VTE topic. For this reason, we have decided to retain it in the manuscript.

6. Please edit for grammar and punctuation.

Response: Thank you for pointing this out. The manuscript has been reread and edited for grammar and punctuation by three native English speakers.

7. Line 176, page 6: please correct spelling in ICER formula.

Response: Thank you. The spelling error has been corrected (line 166).

8. Why is hypotension a branch point in the decision tree? There are no different event probabilities cited based on the presence of hypotension. Please clarify.

Response: Thank you. In our model, we consider hypotension as an intermediate event that could occur during cesarean section surgery. There is evidence that leg compression has been used to reduce the risk of hypotension. As IPC is sometimes used intraoperatively, the branching of hypotension allows the continued use of IPC postoperatively for VTE prophylaxis to be explored in the correct clinical context. There is a difference in the event probabilities as provided in Table 1, Incidence of hypotension without IPC -> 66.29%, Incidence of hypotension with IPC -> 23.86% (0.36*66.29%). This calculation was carried out directly in the model. To make this clearer in the manuscript, we have added the following text as one of the notes to Table 1: “Relative efficacy of IPC on hypotension was estimated from that of lower limb compression, the event rate of hypotension with IPC was then calculated at 66.29% x 0.36 (RR) = 23.86%”.

---

## [Decision Letter · Decision Letter 1]

13 Jun 2023

Cost-effectiveness of mechanical thromboprophylaxis for cesarean deliveries in Brazil

PONE-D-23-07556R1

Dear Dr. Veloz,

We’re pleased to inform you that your manuscript has been judged scientifically suitable for publication and will be formally accepted for publication once it meets all outstanding technical requirements.

Kind regards,

Antonio Simone Laganà, M.D., Ph.D.

Academic Editor

PLOS ONE

Additional Editor Comments (optional):

The authors performed the required corrections, which were positively evaluated by the reviewers. I am pleased to accept this paper for publication.

Reviewers' comments:

Reviewer's Responses to Questions

**Comments to the Author**

1. If the authors have adequately addressed your comments raised in a previous round of review and you feel that this manuscript is now acceptable for publication, you may indicate that here to bypass the “Comments to the Author” section, enter your conflict of interest statement in the “Confidential to Editor” section, and submit your "Accept" recommendation.

Reviewer #1: All comments have been addressed

Reviewer #2: (No Response)

2. Is the manuscript technically sound, and do the data support the conclusions?

Reviewer #1: Yes

Reviewer #2: (No Response)

3. Has the statistical analysis been performed appropriately and rigorously? 

Reviewer #1: Yes

Reviewer #2: (No Response)

4. Have the authors made all data underlying the findings in their manuscript fully available?

Reviewer #1: Yes

Reviewer #2: (No Response)

5. Is the manuscript presented in an intelligible fashion and written in standard English?

Reviewer #1: Yes

Reviewer #2: (No Response)

6. Review Comments to the Author

Reviewer #1: Dear authors

thank you, all comments have been properly addressed

Congratulations, I would recommend It for publication

Reviewer #2: I carrefully evalutated the revised version of this manuscript. Authors have performed the required changes, iproving significantly the quality of the paper.

7. PLOS authors have the option to publish the peer review history of their article (what does this mean?). If published, this will include your full peer review and any attached files.

Reviewer #1: No

Reviewer #2: No

---

## [Editor Report · Acceptance letter]

20 Jun 2023

PONE-D-23-07556R1 

Cost-effectiveness of mechanical thromboprophylaxis for cesarean deliveries in Brazil 

Dear Dr. Veloz:

I'm pleased to inform you that your manuscript has been deemed suitable for publication in PLOS ONE. Congratulations! Your manuscript is now with our production department. 

Kind regards, 

on behalf of

Dr. Antonio Simone Laganà 

Academic Editor

PLOS ONE